# Charge Transfer and Electron Production in Proton Collisions with Uracil: A Classical and Semiclassical Study

**DOI:** 10.3390/ijms24032172

**Published:** 2023-01-21

**Authors:** Clara Illescas, Luis Méndez, Santiago Bernedo, Ismanuel Rabadán

**Affiliations:** Laboratorio Asociado al CIEMAT de Física Atómica y Molecular en Plasmas de Fusión, Departamento de Química, Módulo 13, Universidad Autónoma de Madrid, 28049 Madrid, Spain

**Keywords:** ion collisions with biomolecules, ionization of uracil by proton impact, charge transfer between proton and uracil

## Abstract

Cross sections for charge transfer and ionization in proton–uracil collisions are studied, for collision energies 0.05<E<2500 keV, using two computational models. At low energies, below 20 keV, the charge transfer total cross section is calculated employing a semiclassical close-coupling expansion in terms of the electronic functions of the supermolecule (H-uracil)${}^{+}$. At energies above 20 keV, a classical-trajectory Monte Carlo method is employed. The cross sections for charge transfer at low energies have not been previously reported and have high values of the order of 40 Å${}^{2}$, and, at the highest energies of the present calculation, they show good agreement with the previous results. The classical-trajectory Monte Carlo calculation provides a charge transfer and electron production cross section in reasonable agreement with the available experiments. The individual molecular orbital contributions to the total electron production and charge transfer cross sections are analyzed in terms of their energies; this permits the extension of the results to other molecular targets, provided the values of the corresponding orbital energies are known.

## 1. Introduction

Ion collisions with bio-molecules constitute the first step of cell damage by ion beams, which is the basis of ion-beam cancer therapy. In particular, collisions between fast protons and nucleobases will eventually cause the breakdown of nucleic acids. The collision of a proton with a uracil (U) molecule initially leads to electron removal via charge transfer (CT) and ionization. In the CT process, one electron is captured by the proton: (1)$$H^{+}+U\to H+U^{+}$$

The two-electron capture process to form $H^{-}$ is unlikely and will not be considered. In general, CT is the dominant process for ion-molecule collisions at energies below 25 keV/u that corresponds to an ion velocity equal to 1 atomic unit (the electron velocity in the first Bohr orbit). At higher energies, the main processes involve the ionization of the molecule with the release of one or more electrons: (2)$$H^{+}+U\to H^{+}+U^{q+}+{qe}^{-}$$

Experiments have often considered the total number of electrons lost in the collision, that will be called electron production (EP). It can be noted that, in general, ionization and CT can take place simultaneously and the positive charge of the uracil cation from the ionizing reactions can be larger than one when the CT process takes place. At the relatively high collision energies of the present work, the time spent by the projectile near the target is small compared to the time scales for vibration and rotation of the uracil, so we shall assume that the nuclear degrees of freedom of the target are frozen and that the possible molecular fragmentation would take place in a post-collisional process.

To our knowledge, only Tabet et al. [1] have reported the absolute CT total cross section at a collision energy of 80 keV. Experiments on reactions (2) include the work of Moretto-Capelle and Le Padellec [2], that measured doubly differential cross sections for EP at energies of 25, 50, and 100 keV. Tabet et al. [1] measured the total cross section for EP at 80 keV. Itoh et al. [3] measured EP total and differential cross sections at 0.5, 1.0, and 2.0 MeV. More recently, Chowdhury et al. [4] have measured total and differential EP cross section at 200 keV.

To our knowledge, no *ab initio* close-coupling calculations for CT in $H^{+}+U$ collisions have been carried out, although Bacchus-Montabonel and coworkers calculated CT cross sections in collisions of C${}^{q+}$ ions with uracil and halouracil targets [5,6,7], showing the importance of the orientation effects in these collisions. The large number of electrons of the uracil molecule hinders the application of *ab initio* methods, particularly as the energy increases and the calculation requires many electronic states. This has motivated the use of additivity rules, where the transition probabilities and cross sections are obtained from those of $H^{+}$ collisions with individual atoms or groups of atoms that constitute the molecule; this approach was successfully applied in previous calculations of electron collisions with bio-molecules [8]. Following this idea, Paredes et al. [9] evaluated total and differential cross sections for EP in proton collisions with several bio-molecules by combining experimental data for $H^{+}$ collisions with a few small molecules. In the same vein, Lüdde et al. [10,11] have employed ion–atom cross sections, evaluated using the so-called basis-generator method, to calculate EP cross sections in ion collisions with bio-molecules. Their calculations for collisions with uracil can be found in reference [11].

Lekadir et al. [12] performed calculations of EP cross sections in collisions of multicharged ions with nucleobases. They employed a simple method based on the over-barrier model where the ionized electron follows classical trajectories in the Coulomb field of the projectile and the recoil target represented by a point charge. Sarkadi [13] carried out a systematic classical-trajectory Monte Carlo (CTMC) treatment of ionization in $H^{+}+U$ collisions, in which the active electron moves in the combination of a Coulomb potential created by the projectile and a multicenter model potential describing the electron interaction with U${}^{+}$. There are also some perturbative calculations: continuum distorted wave–eikonal initial state and the first Born approximation with correct boundary conditions were applied by Galassi et al. [14]. CT total cross sections have been reported by Purkait et al. [15]. Covington et al. [16] carried out simulations of the interaction of protons with uracil applying the time-dependent density-functional theory at energies between 15 and 500 keV. They studied energy transfer and ionization at different impact points, showing the relationship with the electron density. They calculated the ratio between CT and ionization probabilities that showed a good agreement with the experiment of Tabet et al. [1].

The first aim of the present work is to provide CT cross sections at energies below 5 keV by applying a semiclassical method in which the projectile follows rectilinear trajectories with respect to the molecule. The electronic wave function is a solution of the semiclassical equation, analogous to the time dependent Schrödinger equation. To take into account the anisotropy of the system, we have considered a set of trajectory orientations and, along each trajectory, the electronic wave function is expanded in a set of adiabatic functions of the ${\left(H-U\right)}^{+}$ supermolecule, as suggested by Rabadán and Méndez [17], and recently applied to $Sn^{3+}$ collisions with H_2_ [18]. The application of the method involves the calculation of the energies of several electronic states of the supermolecule along each trajectory.

The second objective of our work is to explore the use of a simplified method to calculate EP cross sections at intermediate and high collision energies that could be applied to other molecular targets. In this respect, it is known that the EP cross sections are largely determined by the ionization energies of the target orbitals, but because of the high projectile velocity, the ionization does not strongly depend on the details of the electronic density. The procedure employs the CTMC method. In the CTMC, the electron state is described by a distribution that includes a large set of electron trajectories. In our treatment, the electron trajectories are defined by interaction of the electron with the Coulomb potential created by the proton charge, and a one-center screened Coulomb potential that represents the electron interaction with the target molecule. We associate to each molecular orbital (MO) of the uracil the effective charge in the screened Coulomb potential that yields the corresponding ionization energy.

Atomic units are employed unless otherwise stated.

## 2. Results and Discussion

### 2.1. Semiclassical Results

As an illustration, we present, in Figure 1, the 9 PECs along the trajectory t2d with b=7 $a_{0}$. This is one of the four trajectories considered that run perpendicular to the uracil molecular plane, so the PECs are symmetric with respect to the molecular plane. The figure shows the energy of the entrance channel (EC), which dissociates into H${}^{+}$+U(X A’), with a solid black line; its asymptotic value has been set as the zero of energies. Below the entrance channel, there are 6 CT channels, which we have labeled according to their symmetry in the $C_{s}$ point group of $U^{+}$. Above the entrance channel, we show the energies of two CT channels. The A${}^{\prime }$ and A${}^{\prime \prime }$ channels correspond to CASSCF wavefunctions in which the dominant configuration contains a singly occupied a′ and a″ MO, respectively. On the other hand, the dominant configuration of the 3A′ and 4A″ channels contain three unpaired electrons in $U^{+}$, which means that they represent electronic states produced when the uracil loses one electron and another one is excited to the lowest unoccupied MO. The CT channels with asymptotic energy closer to that of the EC are 3A${}^{\prime \prime }$, 4A${}^{\prime \prime }$, and 3A′. The PECs of these channels show avoided crossings with that of the EC, the most external one located around Z≈5 $a_{0}$.

The qualitative picture along trajectories t3 is similar to that shown in Figure 1, but the PECS are not symmetric with respect to Z=0. For trajectories t1, the system preserves the planar symmetry and, given that the EC is of A${}^{\prime }$ symmetry, only states of this symmetry are needed to account for CT to the hydrogen 1s orbital.

State-selected and total CT cross sections are calculated for the 16 subfamilies of trajectories of Figure 2. Within each family, the CT cross sections are obtained by averaging over the corresponding subfamilies: (3)$$\begin{array}{ccc} \sigma_{t1}\left(v\right) & = & \frac{1}{4}\sum_{i\in a,b,c,d}\sigma_{t1i}\left(v\right)+\sigma_{t1ri}\left(v\right) \end{array}$$(4)$$\begin{array}{ccc} \sigma_{t2}\left(v\right) & = & \frac{1}{4}\sum_{i\in a,b,c,d}\sigma_{t2i}\left(v\right) \end{array}$$(5)$$\begin{array}{ccc} \sigma_{t3}\left(v\right) & = & \frac{1}{2}\sum_{i\in a,b}\sigma_{t3i}\left(v\right)+\sigma_{t3ri}\left(v\right) \end{array}$$
where trajectories t1*i* and t3*i* are integrated from $Z_{-}$ to $Z_{+}$, and t1r*i* and t3r*i* from $Z_{+}$ to $Z_{-}$. A total orientation average cross section is obtained by averaging over the three families: (6)$$\sigma_{\mathrm{av}}\left(v\right)=\frac{1}{3}\sum_{i=1,3}\sigma_{ti}\left(v\right)$$
Cross sections (3)–(6) are presented in Figure 3. We note that they are weakly energy dependent. The orientation-averaged one (6) is in between 20 and 40 Å${}^{2}$, with the larger contributions coming from impact parameters of about 7 $a_{0}$. It is worth pointing out that our cross sections are larger than those calculated in references [5,6,7], which are of the order of a few Å${}^{2}$. These are very low cross sections, taking into account the size of the molecular target and that they correspond to collisions with multicharged ions. The main limitation of the present calculation comes from the neglect of the short distances between the proton and the molecule, where the basis becomes quasi-linearly dependent. This is relevant for trajectories with small *b* (≈3 $a_{0}$) that have a small contribution to the total cross section. Assuming that the CT probability is equal to one for b<3 $a_{0}$, the cross section corresponding to those trajectories account for about 10 Å${}^{2}$, which is an upper bound to the uncertainty due to this limitation.

The branching-ratios for the production of different uracil cation states are shown in Figure 4. Trajectory family t1 only contributes to the production of A${}^{\prime }$ cation states when the electron is captured into the H(1s) orbital. The results show that, for collisional energies between 0.5 and 30 keV, the dominant channel is the production of $U^{+}$ in the 2A${}^{\prime }$ state, with values that go from 15 to 20%, while below 0.3 keV dominates the production of 3A${}^{\prime \prime }$ and 4A${}^{\prime \prime }$. The branching ratio to produce the ground state of the cation (1A${}^{\prime \prime }$) is below 5%, as a consequence of the relative large energy gap with the energy of the EC that precludes transitions at low collision energies and the fact that A${}^{\prime \prime }$ states are not populated in t1 trajectories.

Figure 5 displays the molecular orbitals of uracil that are singly occupied in the dominant configurations of the CASSCF wavefuctions of different CT channels (see Table 1). They serve as a guide to pinpoint the bond (or bonds) that is (are) affected when the uracil looses one electron by the proton interaction. In this way, the branching ratios of Figure 4 indicate that, for example, at collision energies of around 3 keV, 20% of the uracil cations are formed in the state 2A${}^{\prime }$, which corresponds to the removal of a electron from the non-bonding orbitals of the oxygens. On the other hand, at collision energies around 0.1 keV, the most populated states are 4A${}^{\prime \prime }$ and 3A${}^{\prime \prime }$, which might trigger the ring opening due to the lost of a ring-bonding electron.

### 2.2. CTMC Results

We have calculated the cross sections $\sigma_{k}^{\mathrm{EP},\mathrm{SC}}$ defined in Equations (20) and (22), for the 10 highest Hartree–Fock MOs of uracil (20–24 A${}^{\prime }$ and 1–5 A${}^{\prime \prime }$) obtained with the basis set mentioned in Section 3.1. The energies of all the valence Hartree–Fock MOs of uracil are tabulated in Table 2. CTMC EP and CT cross sections corresponding to screened Coulomb potentials with the energy given by these 10 MOs are shown in Figure 6 and Figure 7 with symbols, for several collisional energies. These cross sections $\sigma^{\mathrm{EP},\mathrm{SC}}\left(I_{k},E\right)$ seem to follow an inverse law with respect to the ionization energy of the MO:(7)$$\sigma^{\mathrm{EP},\mathrm{SC}}\left(I_{k},E\right)=a{I_{k}}^{-b}.$$
where the parameters *a* and *b*, for each impact energy *E*, are obtained by least square fitting and given in Table 3. For each collisional energy *E*, the fitted $\sigma^{\mathrm{EP},\mathrm{SC}}\left(I_{k},E\right)$ are plotted in Figure 6 and Figure 7 as dashed lines. In the case of EP, the exponential parameter *b* is 1 for collision energies where CT is small compared to EP; i.e., E>100 keV. For those energies, the parameter *a* as a function of impact energy, *E*, fits the Bethe model (see [19]):(8)a(E)=(c+dlnE)/E
with c=129.68 Å${}^{2}$ and d=3.37, with the cross sections expressed in Å${}^{2}$, ionization energies in Hartree and collisional energies in keV.

The parameters of Equation (7) depend only on the ionization energies $I_{k}$ and, accordingly, the present fitting can be employed to obtain EP and CT cross sections using any collection of ionization energies of the target molecule. Indeed, in this work, we use Equation (7), along with the values of the fitted parameters *a* and *b*, to obtain $\sigma_{k}^{\mathrm{EP},C}$ of Equation (20) for the MOs of Table 2 whose corresponding cross sections have not been obtained in our CTMC calculation, and the total cross sections for EP and CT of Equation (19), at each collisional energy, include the contributions of both sets of MOs. With respect to the EP cross sections, an estimation of the contribution of the Auger processes to the EP, $E^{\mathrm{EPa}}$, can be obtained by adding the contribution of the orbitals whose ionization energy is larger than the energy threshold of the process. Using (19), the EP cross section becomes:(9)$$\sigma^{\mathrm{EPa}}\left(E\right)=\sum_{I_{k}>I_{\mathrm{thres}}}\sigma_{k}^{\mathrm{EP}}\left(E\right),$$
where $I_{\mathrm{thres}}$ is taken as the sum of the first and second ionization potentials of the molecule.

We display in Figure 8 the total cross sections for EP calculated using the one-center CTMC method compared with previous experimental and theoretical results. Tabet et al. [1] reported an EP cross section of $177\times 10^{-16}$ cm${}^{2}$ at E=80 keV, which is not included in the figure. One can note a reasonable agreement of all calculations for E≳200 keV and with the experiment of Itoh et al. [3] and Chowdhury et al. [4]. On the other hand, there is a difference between the slopes of the two CTMC calculations that may be due to the one-center and the multicenter model potentials employed. The differences between the results near the maximum of the cross section are very large. As expected, the calculations based on the use of perturbative methods are not appropriate to calculate the total cross section at relatively low energies, but the discrepancy between the shape of the energy dependency of the cross sections of Lüdde et al. [11] and those of Sarkadi [13] is noticeable; it can be noted that the position of the maxima ($E_{\max}\approx 60$ keV) in the works of Lüdde et al. [11] and Paredes et al. [9] is slightly at higher energies than in the present calculation ($E_{\max}\approx 40$ keV) and that of Sarkadi [13]. However, our cross section maximum is higher than those from previous calculations, except that of Lekadir et al. [12] that seems to not fall at the lower energy range. With respect to the convergence of the total cross sections with the number of orbital energies included in the CTMC calculation, we present the results obtained with the 10 highest MO energies and that completed up to 21 MOs, the contribution of the 11 additional MO being about 30% of the first 10 MOs. We also include in the figure the estimate of the Auger process with a gray shade above the 21 MO line.

Our results for CT using the CTMC method are shown in Figure 9 with solid lines, again separating the contribution of the first 10 MO and the total result with 21 MOs. These results are compared to the calculations of Lüdde et al. [20] and Purkait et al. [15], and our own orientation-averaged semiclassical results presented in Figure 3. While a good agreement is found with Lüdde et al. [20] above 50 keV, the CTMC calculations seem to overestimate the capture cross sections below 50 keV, where both our semiclassical calculations and those of Lüdde et al. [10] agree around values of 30 to 40 Å${}^{2}$. Results presented by Purkait et al. [15] are higher than ours by a factor of 3 or 4 at all energies.

## 3. Materials and Methods

### 3.1. Semiclassical Method

In the semiclassical calculation, the projectile follows rectilinear trajectories R=b+vt, where R is the position of the proton with respect to the center of masses of the target molecule, b is the impact parameter and v the collision velocity. We assume that the uracil nuclei remain fixed at their equilibrium positions during the collision (Franck–Condon approximation), and we have considered the lactam tautomer, which is the most stable in gas phase (see [21]). The electronic motion is treated quantum mechanically, and, for each projectile trajectory, the collision wave function Ψ(r,t) is a solution of the eikonal equation: (10)$$\left[H_{\mathrm{el}}-i\frac{\partial }{\partial t}\right]\Psi \left(r,t\right)=0,$$
where r denotes the coordinates of all the electrons of the system and $H_{\mathrm{el}}$ is the clamped-nuclei electronic Hamiltonian of the Born–Oppenheimer approximation for the ${\left(H+U\right)}^{+}$ system. The wave function Ψ is expanded in a basis of approximate eigenfunctions of $H_{\mathrm{el}}$: (11)$$\Psi \left(r,t\right)=D\left(r,t\right)\sum_{k}c_{k}\left(t\right)\Phi_{k}\left(r;R\right)\exp\left(-i\int_{0}^{t}E_{k}dt\right)$$
with
(12)$$H_{\mathrm{el}}\Phi_{k}\left(r;R\right)=E_{k}\left(R\right)\Phi_{k}\left(r;R\right),$$
and *D* is a common translation factor [22]. The substitution of (11) into (10) yields a set of first order differential equations for the coefficients $c_{k}$, which lead to the transition probabilities and cross sections.

In order to obtain orientation-averaged cross section, the calculations are carried out for a set of molecular orientations or, alternatively, for a set of trajectory orientations with respect to a fixed molecule, as illustrated in Figure 2. The present calculation is based on the average procedure of Errea et al. [23] that employs a 6-point Cotes formula for the velocity orientations; for each orientation of v, we consider four orientations of b. The trajectories related by reflection through the molecular plane are equivalent, which results in the 16 trajectory orientations, sketched in Figure 2.

The electronic wavefunctions $\Phi_{k}\left(r;R\right)$ have been calculated *ab initio* using the complete-active-space self-consistent-field (CASSCF) method with MOLPRO [24] and the triple-zeta basis set of Widmark et al. [25] on the C, N, and O atoms, while a double-zeta one is located on the H atoms. In the CASSCF calculation, the first 23 MOs are doubly occupied, while the following 8 hold 12 electrons. In total, 9 electronic states are obtained with equal weight as the optimization prescription. For each subfamily of trajectories, the electronic wavefunctions are obtained at a set of ion positions {$b_{i},Z_{j}=vt_{j}$}, where $b_{i}\in \left[4.0,12.0\right]$ $a_{0}$ and $Z_{j}\in \left[-20.0,+20.0\right]$ $a_{0}$. The wavefunctions are also obtained at the geometries {$b_{i},Z_{j}+\delta$}, with δ=0.0001 $a_{0}$, to calculate the non-adiabatic couplings, using a two-point numerical differentiation method of MOLPRO. About 10 values of $b_{i}$ are considered, whose range depends on the trajectory subfamily, and about 300 values of $Z_{i}$, whose range depends on the value of $b_{i}$. In summary, we perform about 6000 CASSCF calculations for each trajectory subfamily. Then, for each trajectory ($b_{i},\hat{v}$), we obtain 9 potential energy curves (PECs), $E_{k}\left(Z_{i}\right)$, corresponding to the 9 electronic states included in the expansion (11), 36 non-adiabatic matrix elements, $A_{kk^{\prime }}\left(Z_{i}\right)$, with ($k=1,9;k^{\prime }=k+1,9$), and the double moments $XX_{kk^{\prime }}\left(Z_{i}\right)$, $XZ_{kk^{\prime }}\left(Z_{i}\right)$, and $ZZ_{kk^{\prime }}\left(Z_{i}\right)$ to account for the translation factor corrections derived from the switching function of Errea et al. [26].

### 3.2. CTMC Method

As in the semiclassical calculation, the eikonal-CTMC applied in this calculation assumes that the ion follows rectilinear trajectories with respect to the target molecule. The application of the CTMC method [27] to ion–molecule collisions with a one-center model potential was shown by Errea et al. [28]. In this approach, the motion of the active electron is described by means of a classical distribution function, ρ(r,p,t;v,b), which is discretized by using a set of electronic ($10^{6}$ in the present calculation) independent trajectories. In these trajectories, the electron moves in the field created by the projectile and a screened Coulomb potential with effective charge $Q_{k}$ that approximates the interaction with the molecular core; this yields the Hamiltonian: (13)$$h_{k}=\frac{p^{2}}{2}-\frac{Q_{k}}{r_{t}}-\frac{1}{r_{p}},$$
where p is the electronic momentum and $r_{p}$ and $r_{t}$ are the electronic position vectors with respect to the projectile and the target, respectively.

The effective charges $Q_{k}$ are obtained by fitting the ionization energies, $I_{k}$, of the molecular orbitals: (14)$$Q_{k}=\sqrt{2n^{2}I_{k}},$$
with n=2 in the present calculation. For each value of $Q_{k}$, a CTMC calculation is carried out yielding the CT and ionization one-electron probabilities $p_{k}^{\mathrm{cap}}$, $p_{k}^{\mathrm{ion}}$ for removing the electron from the corresponding molecular orbital, and $p_{k}^{\mathrm{el}}=1-p_{k}^{\mathrm{cap}}-p_{k}^{\mathrm{ion}}$ is the probability that the electron remains attached to the target. In our treatment, the initial electron distribution $\rho_{k}\left(r,p,t\to -\infty \right)$ is a hydrogenic one [29] built with a linear combination of 7 microcanonical distributions, $\rho_{k}^{j}$, of different energies $E_{j}$: (15)$$\rho_{k}\left(r,p,t\to -\infty \right)=\sum_{j=1}^{7}a_{k}^{j}\rho_{k}^{j}\left(r,p;E_{j}\right),$$
where the coefficients $a_{k}^{j}$ have been chosen by fitting the quantal position and momentum distributions, and checked that $\sum_{j}a_{k}^{j}E_{j}≃E_{k}$. When using the hydrogenic distribution, the total EP cross section is compatible with the Bethe limit, as shown by Illescas and Riera [30].

The many-electron transition probabilities are obtained from those of the one-electron calculation by applying the standard independent particle model (IPM) (see reference [31]). For instance, for a set of *N* doubly occupied orbitals, the probabilities for single ionization are: (16)$$P^{\mathrm{ion}}=\sum_{k=1}^{N}2p_{k}^{\mathrm{ion}}p_{k}^{\mathrm{el}},$$
and, similarly, for single CT: (17)$$P^{\mathrm{sc}}=\sum_{k=1}^{N}2p_{k}^{\mathrm{cap}}p^{\mathrm{el}}.$$
The probability for EP and net capture are:(18)$$P^{\mathrm{EP}}=\sum_{k=1}^{N}2p_{k}^{\mathrm{ion}}\;;\;P^{C}=\sum_{k=1}^{N}2p_{k}^{\mathrm{cap}}.$$
It is useful to express the total cross sections for these inclusive processes in the form
(19)$$\sigma^{\mathrm{EP}}=\sum_{k=1}^{N}\sigma_{k}^{\mathrm{EP}}\;;\;\sigma^{C}=\sum_{k=1}^{N}\sigma_{k}^{C},$$
where
(20)$$\begin{array}{c} \sigma_{k}^{\mathrm{EP}}=2\pi \int_{0}^{\infty }2p_{k}^{\mathrm{ion}}bdb \end{array}$$
(21)$$\begin{array}{c} \sigma_{k}^{C}=2\pi \int_{0}^{\infty }2p_{k}^{\mathrm{cap}}bdb, \end{array}$$
and analogously, for single CT:(22)$$\sigma_{k}^{\mathrm{SC}}=2\pi \int_{0}^{\infty }2p_{k}^{\mathrm{cap}}p_{k}^{\mathrm{el}}bdb.$$

## 4. Conclusions

In this paper we have calculated cross sections for charge transfer and electron production in proton collisions with uracil. At low energies, we have carried out a semiclassical close-coupling calculation in terms of *ab initio* wave functions. The orientation averaged total cross section for charge transfer is almost constant in the energy range 0.05<E<50 keV with values of the order of 30 Å${}^{2}$, in contrast to the cross of ≈3 Å${}^{2}$ reported previously for collisions of C${}^{4+}$ with uracil. The state-selective cross sections indicate that the main exit capture channel involve the removal of one electron from non-bonding orbitals, which points toward a low fragmentation probability of the uracil cation after the charge transfer process.

Electron production and charge transfer cross sections have also been calculated with a CTMC method for energies above 20 keV and up to 2500 keV and 250 keV, respectively. These calculations required consideration of the contributions of all the valence molecular orbitals of uracil to the processes. We have performed the full CTMC calculation with the highest 10 molecular orbitals of uracil, and we have used those data to fit the cross sections to an inverse law of the molecular orbital energy, for each impact energy value. This allows us to obtain the corresponding cross sections for the molecular orbitals that have not been treated explicitly with the CTMC model and, with those data, we have carried out the calculation of the total electron production and charge transfer cross sections. These results are in good agreement with previous experimental and theoretical results. We have fitted the parameters of an inverse law, and they could be used to obtain the EP and CT cross sections from any set of molecular orbital energies or ionization potential energies. This method can be used to estimate electron production and charge transfer cross sections in proton collisions with other molecules.

## Figures and Tables

**Figure 1 ijms-24-02172-f001:**
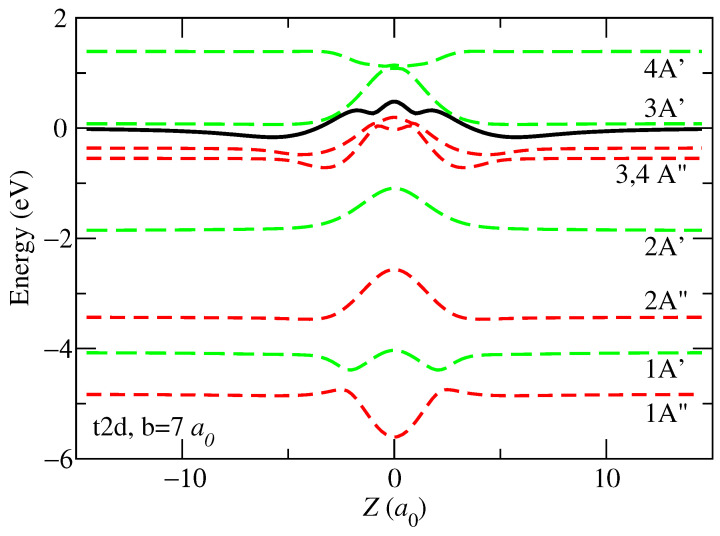
Potential energy curves along the trajectory t2d with impact parameter b=7 $a_{0}$. The states are labeled according to the electronic state of $U^{+}$ when the projectile is at an asymptotic distance. The energy curve of the entrance channel, corresponding to $H^{+}$+uracil(X A′), is shown with a black solid line. The CT channels are shown with dashed lines.

**Figure 2 ijms-24-02172-f002:**
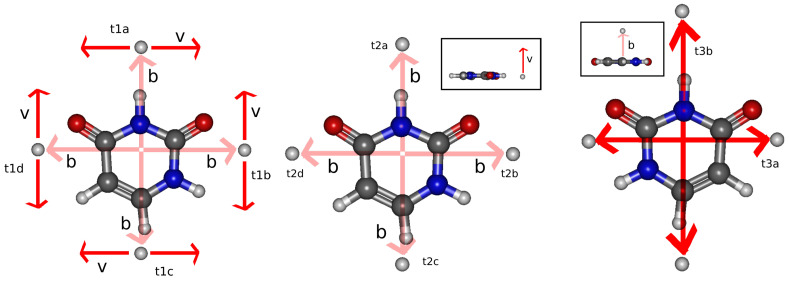
Projectile trajectories characterized by the set {$\hat{b}$, $\hat{v}$}. Taking into account the planar geometry of the molecule, the 24 trajectories of a 6-point Cotes formula are reduced to the 16 trajectories represented in the figure. The trajectories are classified in families attending to whether $\hat{v}$ is perpendicular (t1), contained in (t2) or parallel to (t3) the molecular plane, and subfamilies t1x, t2x, and t3x that share the same unitary vectors ($\hat{v},\hat{b}$).

**Figure 3 ijms-24-02172-f003:**
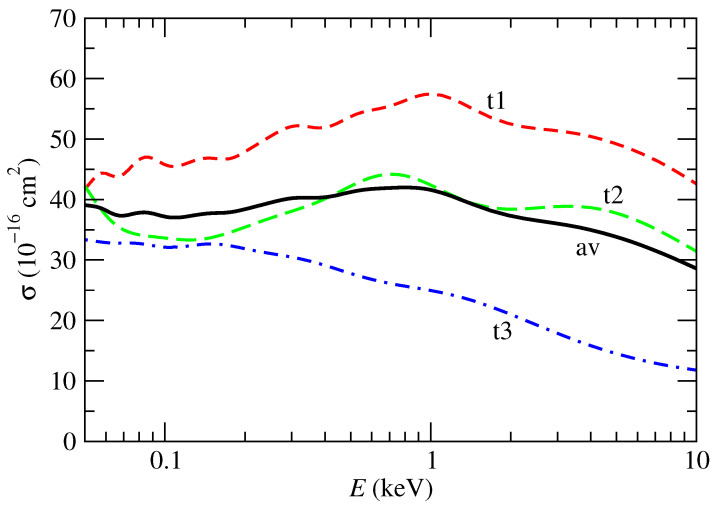
Total single charge transfer cross section in collisions of protons with uracil molecules as functions of the collision energy. Broken lines correspond to the averages of the subfamilies within a family, Equations (3)–(5), labeled in the figure; the solid line is the average of the three families and corresponds to the orientation average of Equation (6).

**Figure 4 ijms-24-02172-f004:**
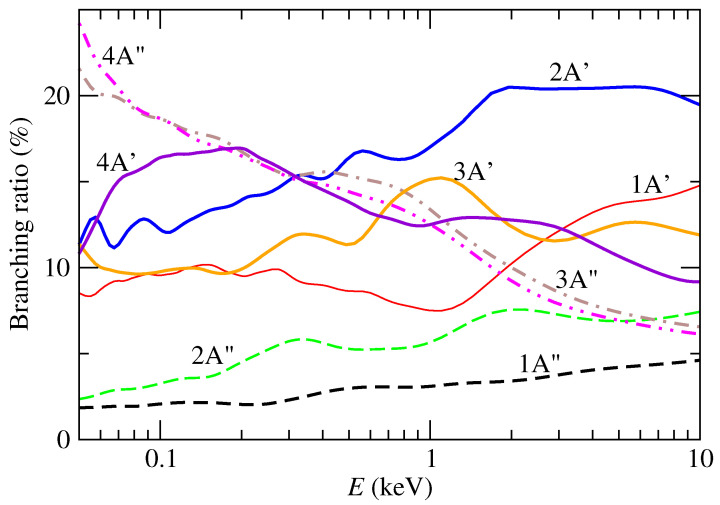
Branching-ratio for production of uracil cations by CT in collisions of protons with uracil molecules. Broken lines correspond to ions in A${}^{\prime \prime }$ state, while solid ones are those in a A${}^{\prime }$ state.

**Figure 5 ijms-24-02172-f005:**
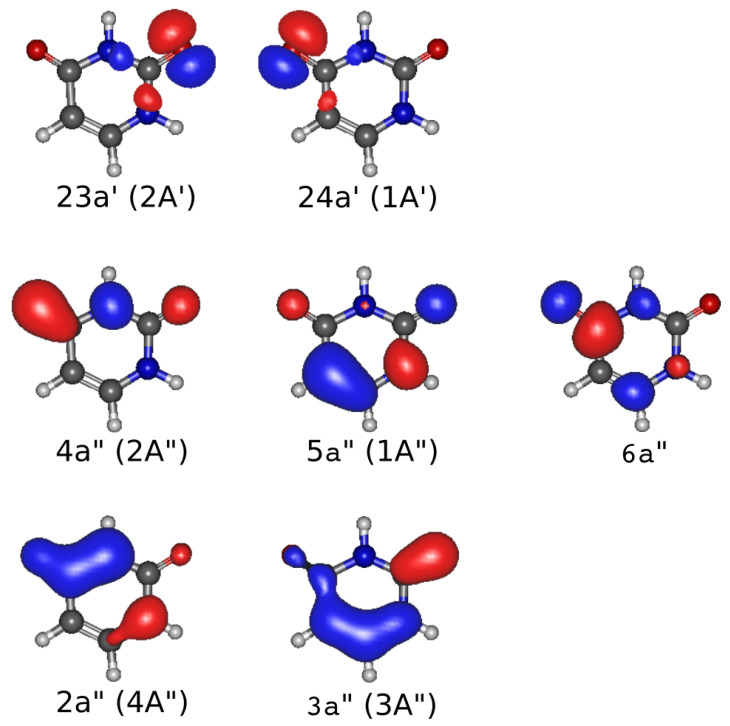
Natural orbitals of the uracil cation obtained with the CASSCF calculations when the projectile is at asymptotic distances along the t2d trajectory. The labels underneath each orbital refer to the sequence number of the orbital within its symmetry in the Cs point group, and the bracket contains the $U^{+}$ electronic state of the uracil cation (see Table 1).

**Figure 6 ijms-24-02172-f006:**
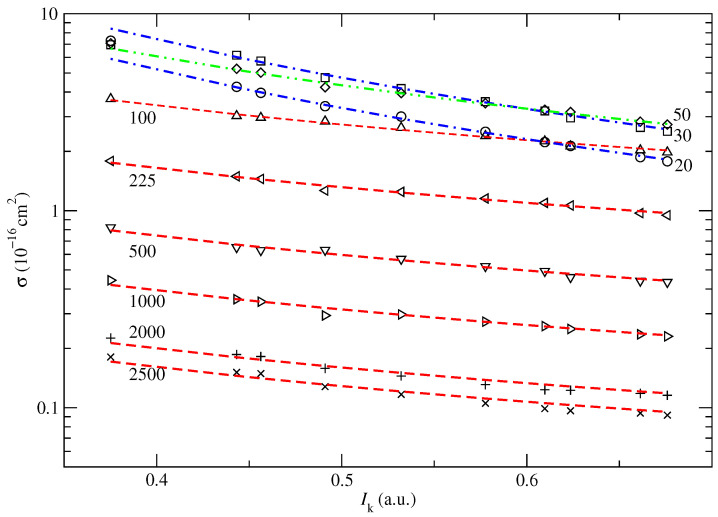
Electron production cross sections from individual MO of uracil after collision with protons, given by their ionization energy $I_{k}$. The different symbols correspond to different collision energies specified in the figure with numbers in keV. The lines are the fitted Equation (7).

**Figure 7 ijms-24-02172-f007:**
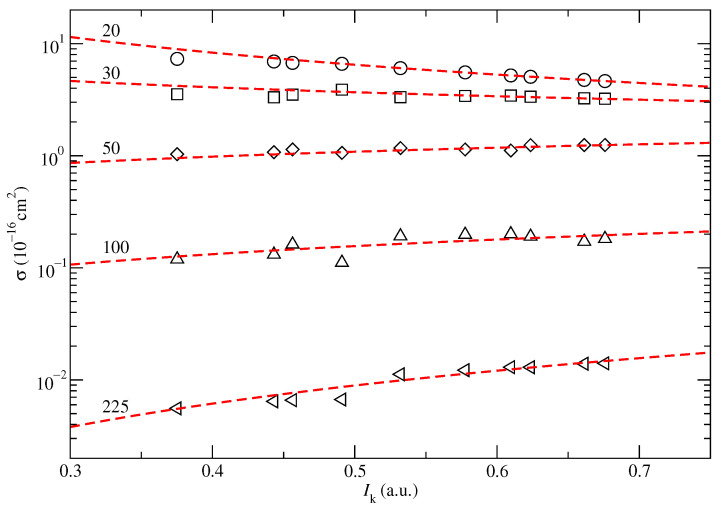
Same as in Figure 6, but for the single charge transfer process.

**Figure 8 ijms-24-02172-f008:**
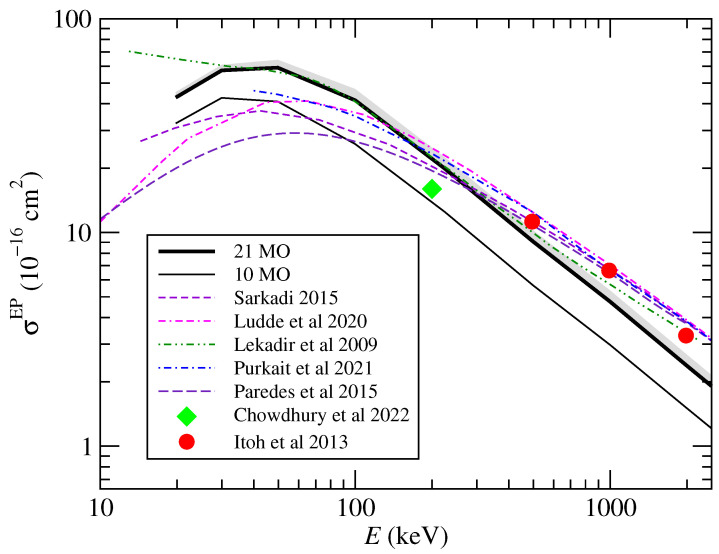
Total cross sections for electron production in proton–uracil collisions as functions of the collision energy. The present CTMC results (solid lines marked with 21 MO and 10 MO) are compared with those of previous calculations Paredes et al. [9], Lüdde et al. [11], Lekadir et al. [12], Sarkadi [13], and the experimental results of Itoh et al. [3] and Chowdhury et al. [4],as indicated in the figure. An estimation of the Auger contribution to ionization is added with a gray shade.

**Figure 9 ijms-24-02172-f009:**
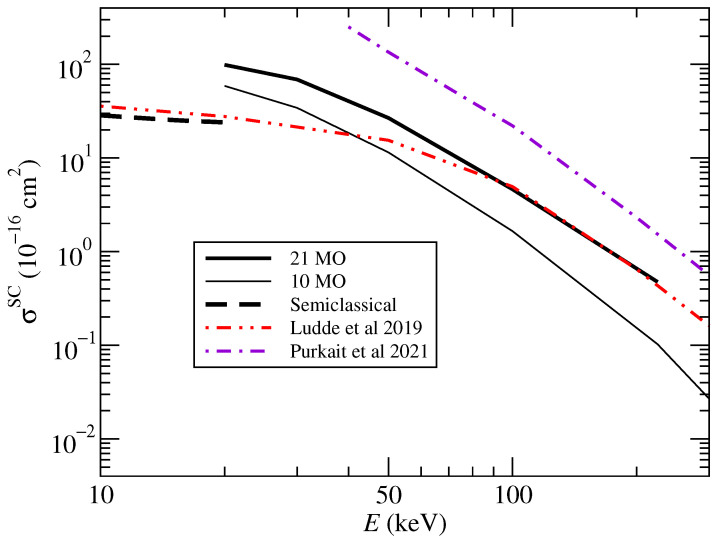
Cross sections for single charge transfer in proton–uracil collisions as functions of the collision energy. The present CTMC results (21 MO and 10 MO) are compared with those of Lüdde et al. [20] and Purkait et al. [15] and with the semiclassical results shown in Figure 3.

**Table 1 ijms-24-02172-t001:** Electronic states of $U^{+}$ with specification of the main electronic configuration (see molecular orbitals in Figure 5) and the electronic energies (in eV) referred to its ground state.

Electronic State	Dominant Configuration	Energy (eV)
1A${}^{\prime \prime }$	${\left(23a^{\prime }\right)}^{2}{\left(2a^{\prime \prime }\right)}^{2}{\left(3a^{\prime \prime }\right)}^{2}{\left(4a^{\prime \prime }\right)}^{2}{\left(24a^{\prime }\right)}^{2}{\left(5a^{\prime \prime }\right)}^{1}$	0.0
1A${}^{\prime }$	${\left(23a^{\prime }\right)}^{2}{\left(2a^{\prime \prime }\right)}^{2}{\left(3a^{\prime \prime }\right)}^{2}{\left(4a^{\prime \prime }\right)}^{2}{\left(24a^{\prime }\right)}^{1}{\left(5a^{\prime \prime }\right)}^{2}$	0.75
2A${}^{\prime \prime }$	${\left(23a^{\prime }\right)}^{2}{\left(2a^{\prime \prime }\right)}^{2}{\left(3a^{\prime \prime }\right)}^{2}{\left(4a^{\prime \prime }\right)}^{1}{\left(24a^{\prime }\right)}^{2}{\left(5a^{\prime \prime }\right)}^{2}$	1.40
2A${}^{\prime }$	${\left(23a^{\prime }\right)}^{1}{\left(2a^{\prime \prime }\right)}^{2}{\left(3a^{\prime \prime }\right)}^{2}{\left(4a^{\prime \prime }\right)}^{2}{\left(24a^{\prime }\right)}^{2}{\left(5a^{\prime \prime }\right)}^{2}$	2.95
3A${}^{\prime \prime }$	${\left(23a^{\prime }\right)}^{2}{\left(2a^{\prime \prime }\right)}^{2}{\left(3a^{\prime \prime }\right)}^{1}{\left(4a^{\prime \prime }\right)}^{2}{\left(24a^{\prime }\right)}^{2}{\left(5a^{\prime \prime }\right)}^{2}$	4.28
4A${}^{\prime \prime }$	${\left(23a^{\prime }\right)}^{2}{\left(2a^{\prime \prime }\right)}^{1}{\left(3a^{\prime \prime }\right)}^{2}{\left(4a^{\prime \prime }\right)}^{2}{\left(24a^{\prime }\right)}^{2}{\left(5a^{\prime \prime }\right)}^{2}$	4.47
3A${}^{\prime }$	${\left(23a^{\prime }\right)}^{2}{\left(2a^{\prime \prime }\right)}^{2}{\left(3a^{\prime \prime }\right)}^{2}{\left(4a^{\prime \prime }\right)}^{2}{\left(24a^{\prime }\right)}^{1}{\left(5a^{\prime \prime }\right)}^{1}{\left(6a^{\prime \prime }\right)}^{1}$	4.91
4A${}^{\prime }$	${\left(23a^{\prime }\right)}^{2}{\left(2a^{\prime \prime }\right)}^{2}{\left(3a^{\prime \prime }\right)}^{2}{\left(4a^{\prime \prime }\right)}^{2}{\left(24a^{\prime }\right)}^{1}{\left(5a^{\prime \prime }\right)}^{1}{\left(6a^{\prime \prime }\right)}^{1}$	6.22

**Table 2 ijms-24-02172-t002:** Molecular orbital ionization energies (in Hartree) of uracil at the Hartree–Fock level, $I_{k}$. A${}^{\prime }$ MO from 20 to 24 and all A${}^{\prime \prime }$ are used in the CTMC calculations, while A${}^{\prime }$ orbitals from 9 to 19 are only used to compute the final CTMC cross sections.

MO (A${}^{\prime }$)	$I_{k}$	MO (A${}^{\prime \prime }$)	$I_{k}$
24	0.4563	5	0.3752
23	0.4910	4	0.4432
22	0.6097	3	0.5321
21	0.6236	2	0.5776
20	0.6613	1	0.6759
19	0.6860		
18	0.7224		
17	0.7742		
16	0.8153		
15	0.9109		
14	0.9416		
13	1.0954		
12	1.2528		
11	1.3211		
10	1.4068		
9	1.4442		

**Table 3 ijms-24-02172-t003:** Parameters of Equation (7) for the fit of the electron production and charge transfer at a given collision energy as a function of the ionization potential of the first 10 MOs (20–24 A${}^{\prime }$ and 1–5 A${}^{\prime \prime }$, see Table 2) of uracil.

	Electron Production		Charge Transfer	
*E* (keV)	*a*	*b*	*a*	*b*
20	0.8298	2.0	3.557	0.790
30	1.1820	2.0	3.145	0.138
50	1.5294	1.5	1.385	−0.305
100	1.3643	1.0	0.266	−0.772
225	0.6584	1.0	0.030	−1.815
500	0.2993	1.0		
1000	0.1571	1.0		
2000	0.0769	1.0		
2500	0.0615	1.0		

## Data Availability

Not applicable.

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
