# Peer review of "Charge Transfer and Electron Production in Proton Collisions with Uracil: A Classical and Semiclassical Study"

_ijms, 2023, doi:10.3390/ijms24032172_

Round 1

Reviewer 1 Report

Charge transfer and electron production in proton collisions with uracil. A classical and semiclassical study

Clara Illescas * , L Mendez , S Bernedo , I. Rabadán *

  In the present manuscript, the authors address a study of charge transfer and electron production in collisions between protons and uracil molecules over a large energy range. The methods used differ depending on the impact energy. For energies smaller than 20 keV, the authors use a semiclassical molecular close-coupling method, and for energies larger than 20 keV, they apply a classical -trajectory Monte Carlo method. With both methods, the authors have documented their long experience in many publications.   The manuscript is well suited for publication in International Journal of Molecular Sciences. The presentation of the methods used is scientifically clear and comprehensible, and the results are well documented. The manuscript can be accepted provided the authors comment on the following item:   ·      In the discussion of Figs. 8 and 9 and/or in the conclusion, respectively, I miss a comment on why the results for EP as well as CT seem to fall off more steeply at large energies than the cited results from the literature.

Author Response

We thank very much for the comments. The manuscript can be accepted provided the authors comment on the following item:
In the discussion of Figs 8 and 9 and/or in the the conclusion, respectively,
I miss a comment on why the results for EP as well as CT seem to fall off more steeply
at large energies that the cited results from the literature.
We have included a comment for EP in the discussion of Fig. 8, on line 230. "On the other hand, there is a difference between the slopes of the two CTMC calculations that
may be due to the one-center and the multicenter model potentials employed." We have modified Fig. 9 by shortening the energy range to 300 keV, in accordance with the
highest energy (fitted) showed in Fig. 7. The fit could be not good enough for E > 225 keV due to the uncertainty in the calculation
of the (very small) cross sections at 500 keV. We have included a new reference [31] that was mislabeled.

Reviewer 2 Report

  In the present manuscript "Charge transfer and electron production in proton collisions with uracil. A classical and semiclassical study", by Illescas et al. two methods i.e.  semiclassical approximation and Monte Carlo are employed to compute cross-collision sections of proton-uracil encounters. Overall, I found this manuscript's main ideas and results exciting and valuable for the research community. Monte Carlo method can lead to a more efficient procedure for the calculation of CCS as compared to the experiment for instance. The methodology is well described, and the conclusions are consistent with the arguments presented. For these reasons, I would accept this article in its present form.
I have some additional questions to the authors though, I think you are assuming that the uracil is in a fixed geometry, how could the flexibility of this molecule could affect the results?  
Another thing, it seems that you have compiled plenty of data and the analysis that you tried is a regression model but would
machine learning/deep learning be better options in the present case?
Also, you have used the Monte Carlo method only for the high energy range, but could you use it
for the entire range? Would it be more/less efficient/accurate? 

Author Response

We thank very much for the comments, which we address in the following answers:
How could the flexibility of this molecule affect the results? At the energy range explored, the interaction time of the projectile with the molecule is short
enough to consider frozen the vibrational and rotational degrees of freedom of the molecule.
On the other side, allowing for different geometries around the equilibrium one would slightly change
the energy of the molecular orbitals used in the CTMC calculations and the electronic energies used
in the semiclassical ones: for some of the geometries the cross sections would be slightly higher,
for some slightly lower and the final geometry average result tend to be very close to the one obtained with the single equilibrium geometry. We could say that the uncertainties in the cross sections due to the approximation of using a single equilibrium geometry are small compared to the absolute value of the cross sections. Would machine learning/deep learning be better options in the present case?
The regressions performed in this work rely on very few parameters of simple models, so we think that,
in this case, the human-guided fitting is a better option. You have used the Monte Carlo Method only for the high energy range, but could you use it for the entire range? CTMC methods are adept to simulate interactions when quantum effects are negligible,
or the system can be reduced to a small number of degrees of freedom.
In this case, the system has a considerable number of electrons and some electronic states involved in
the electron capture process involved the simultaneous capture and excitation of the target (two electrons)
, so a semiclassical treatment involving electronic wavefunction is adecuate.
